# Optimization and Benefit Analysis of Grain Trade in Belt and Road Countries

**DOI:** 10.3390/e24111667

**Published:** 2022-11-15

**Authors:** Ruijin Du, Yang Chen, Gaogao Dong, Lixin Tian, Jing Zhang, Nidan Zhang

**Affiliations:** 1Center of Energy Development and Environmental Protection, Jiangsu University, Zhenjiang 212013, China; 2School of Mathematical Sciences, Jiangsu University, Zhenjiang 212013, China; 3Emergency Management Institute, Jiangsu University, Zhenjiang 212013, China

**Keywords:** opportunity cost, resource benefit, reconstruction of grain circulation pattern, linear programming optimization model, Belt and Road economies

## Abstract

Grain trade in Belt and Road (B&R) countries shows a mismatch between the volume and direction of grain flows and actual demand. With economic and industrial development, the water crisis has intensified, which poses a great challenge to the security of world grain supply and demand. There are few studies on the reconstruction of grain trade relations from the perspective of grain economic value. In this paper, a linear optimization model considering opportunity cost is proposed to fill the gap, and it is compared and analyzed with the optimization model considering only transportation cost. The grain supply and demand structures in both optimization results show characteristics of geographical proximity and long-tail distribution. Furthermore, the economic and water resource benefits resulting from the two optimal configurations are compared and analyzed. It is found that the economic benefits generated by grain trade in B&R countries with the consideration of opportunity cost not only cover transportation costs but also generate an economic value of about 130 trillion US dollars. Therefore, considering opportunity cost in grain trade is of great significance for strengthening cooperation and promoting the economic development of countries under the B&R framework. In terms of resource benefits, the grain trade with consideration of opportunity cost saves nearly 28 billion cubic meters of water, or about 5% of the total virtual water flow. However, about 72 billion cubic meters of water is lost for the grain trade with consideration of transportation cost. This study will help to formulate and adjust policies related to the “Belt and Road Initiative” (B&R Initiative), so as to maximize the economic benefits while optimizing the structure of grain trade and alleviating water scarcity pressures.

## 1. Introduction

The B&R Initiative, proposed by China in 2013, has drawn widespread political and academic attention. The inter-regional trade in goods and services along the B&R has positive effects on economic integration [1], the globalization of medical and health industries [2], inter-regional connectivity, industrial integration [3] and the sustainable development of resources [4,5]. While people gain economic benefits from trade, they are increasingly aware that the associated environmental problems have brought serious threats to human existence. A large number of existing studies have focused on the impact of trade on carbon emissions [6,7,8,9], while less attention has been paid to the impact on water resources. Meeting the growing water demands of ecosystems and society is one of the major environmental challenges in the 21st century.

On the one hand, as the impact of climate change and environmental pollution intensifies, the availability of water resources decreases dramatically [10,11]. On the other hand, factors such as population growth and socio-economic development have led to increasing global water demand [12,13]. Under the action of dual pressures, the situation of water resources in the world has become more severe. According to the 2020 United Nations State of Food and Agriculture report, 3.2 billion people worldwide face water scarcity, and about 1.2 billion people live in agricultural areas with extreme water scarcity [14]. Agriculture remains the largest user of water consumption globally, accounting for 87% of global water consumption [15]. Grain production, however, is an important component of agricultural water consumption and plays an important role in economic development, water resources utilization, and social stability [16]. Virtual water is not water in the real sense, but the water resources needed to produce products and services, that is, virtual water condensed in products and services. This concept was first proposed by Tony Allan, which provides a new perspective for the research of water resources security and management [17,18,19,20]. Compared with physical water resources, its easy transportation characteristics make virtual water trade a useful tool for mitigating water scarcity [21,22]. In this paper, virtual water refers to water resources that are embodied in grain in an “invisible” form. The virtual water strategy combines grain production and water use, aiming at a global reallocation of water resources by linking grain availability in water-rich areas with improved water scarcity in water-scarce areas through trade. In this work, the virtual water strategy is defined as the import of grain from water-scarce countries or regions to water-abundant countries or regions through trade in order to alleviate global grain supply and demand constraints and water scarcity.

In accounting for virtual water flows, there are mainly input–output models and linear optimization models. Some studies have focused on the use of input–output (IO) analysis to reveal virtual water in a country or region [23,24]. In the case of unbalanced regional production and consumption structures, it becomes crucial to assess the virtual water flows between regions. The multi-regional input–output (MRIO) model can systematically describe the input–output relationships between different sectors in different regions, and it has been widely used to calculate the virtual water flows between regions. For example, Zhang et al. [25] used the 2012 Chinese MRIO table to account the virtual water volume traded between the Yellow River Delta and other provinces, and to assess the dependence of the Yellow River Delta on external water resources. The results illustrate that virtual water trade exacerbates water scarcity in this region, as virtual water exports are greater than imports. Zhang et al. constructed a virtual water trade network based on the MRIO model and accounted for virtual scarce water in sectoral exports of intermediate and final products to study the virtual water flow risks by sectors in northeast China [26]. An et al. [27] quantified the virtual water flow embodied in the inter-provincial grain trade in China, which was validated with the results calculated by MRIO. Several shortcomings have been revealed in the application of input–output models to address issues related to virtual water embodied in grain trade. First, the input–output model considers the economic sector level rather than individual product level [28], making it difficult to accurately assess international grain trade based on these sectors. Secondly, the time resolution of the input–output table is poor, and there exists a time delay [29]. Currently, the open-source data of the input–output table in the Eora database is only updated to 2016. The linear optimization model can compensate for the above shortcomings of the input–output model. Detailed supply and demand structures of grain products among countries can be obtained from the latest grain-related data. Linear programming is an important system optimization method in operations research, providing a scientific basis for making optimal decisions with limited human, material, financial and other resources in a rational way. From the perspective of virtual water, international grain trade shows an irrational structure of ”North-to-South Water Diversion” [30,31,32]. The spatial dislocation of grain production and water poses a major challenge to sustainable development. The trade pattern of grain, coupled with the scarcity and endowment difference of water resources, has seriously threatened the development of agriculture. Therefore, reshaping the grain trade relations has positive significance for solving the water crisis [33].

In addition, when weighing the rationality of the grain trade structure, it is mostly the transportation costs that are considered [27,34,35]. Grains are used not only to meet ration consumption but also to meet the needs of industry and economic development. Opportunity cost refers to the loss of potential benefits resulting from choosing one better alternative and giving up the other when making a decision [36]. According to the explanation of the opportunity cost (the maximum net income that may be obtained by making a choice but giving up another), it is necessary to explore the role of the opportunity cost of grain in order to stimulate the dynamism of grain in the industrial value chain and to enhance the added value of grain. In general, opportunity cost reflects the value of the best alternative and should be considered as part of any decision-making process. However, the hidden aspects of opportunity cost are difficult to capture and measure, and they are often ignored by people, so that optimal decisions cannot be made [37,38,39,40]. Therefore, when measuring the benefits of different decisions, decision makers should consider not only the resources sacrificed after making the choice but also the potential benefits lost in comparison with other options, that is, the opportunity cost. This study mainly focuses on the added value of industrial uses of grain (other service uses are not considered), using the added value generated per tonne of industrial grain as the opportunity cost, and a linear optimization model of grain trade in countries along the B&R is constructed. With the objective of minimizing the cost of grain trade, the optimal configuration of grain trade and the economic and water benefits in two situations with or without considering the opportunity cost are compared. This exploration helps policy makers understand the impact of grain trade structure on human society.

The rest of the paper is organized as follows: Section 2 introduces the linear optimization model and method for quantifying benefits in two scenarios, and it presents the sample data selected for the study. Section 3 obtains the optimal configuration of grain trade and compares the two optimization results to reveal the impact of opportunity cost on economic and water resource benefits. Eventually, Section 4 draws conclusions on the optimal configuration and benefit analysis of grain trade in the B&R countries, and it puts forward more policy suggestions.

## 2. Data and Methods

### 2.1. Data

Grain-related data (the grain consists of rice, wheat, maize and soybeans, of which the statistical caliber is consistent with the International Statistical Yearbook) of countries along the B&R in 2018 were selected to assess trade flows of grain and construct a grain trade network model. The grain production, grain harvested area, annual output of livestock meat, and annual output of industrial products such as alcohol are all from the Food and Agriculture Organization of the United Nations (https://www.fao.org/statistics/en/ (accessed on 14 November 2022)). The industrial added value and population data are from World Bank public data (https://data.worldbank.org.cn/indicator (accessed on 14 November 2022)). Meteorological data, including maximum temperature, minimum temperature, average temperature, dew point temperature, sunshine hours, wind speed, etc. are from National Centers for Environmental Information (https://www.ncei.noaa.gov (accessed on 14 November 2022)) and European Centre for Medium-Range Weather Forecasts (https://www.ecmwf.int (accessed on 14 November 2022)). The geographic distance data between countries comes from the CPEII database (http://cepii.fr/CEPII/en/welcome.asp (accessed on 14 November 2022)). Grain consumption in this study includes five categories: ration consumption, feed consumption, industrial consumption, seed consumption and grain loss [41]. The details of the specific data processing are shown in Table 1.

Due to missing data, 40 countries along the B&R are considered in this paper, including 23 Asian countries (Armenia, Cambodia, China, Georgia, India, Indonesia, Iran, Israel, Jordan, Kazakhstan, Kyrgyzstan, Laos, Lebanon, Malaysia, Mongolia, Myanmar, Nepal, Sri Lanka, Tajikistan, Thailand, Turkey, Uzbekistan, Vietnam), 15 European countries (Albania, Belarus, Czech Republic, Estonia, France, Hungary, Latvia, Lithuania, Moldova, Poland, Romania, Russia, Slovakia, Slovenia, Ukraine) and 2 African countries (Egypt, Zimbabwe), for which information is listed in Appendix A
Table A1.

### 2.2. Linear Optimization Model for Grain Trade

According to the production and consumption of grains in each country, the 40 countries along the B&R are divided into grain-rich countries and grain-deficit countries, where Equation (Equation 1) quantifies the grain balance of country *i*:(1)$$B_{i}=P_{i}-C_{i}\;\left(i=1,2,\cdots ,n\right),$$
where $P_{i}$ and $C_{i}$ represent the production and consumption of grain in country *i*, respectively. If $B_{i}>0$, country *i* is a grain-rich country and can export grain; otherwise, country *i* is grain-deficit and has to import grain from abroad.

The fundamental position of agriculture in the national economy influences the profitability of the grain trade, and grain logistics is directly related to its trade relations. Grain trade seeks to achieve greater economic benefits by minimizing transportation costs. In this way, a linear optimization Model I with the objective of minimizing transportation cost is constructed to adjust the trade pattern of grain.

Model I: Linear optimization model with consideration of transportation cost

Objective function:(2)$$min\;T=\sum_{i=1}^{n-m}\sum_{j=1}^{m}x_{i,j}\times c_{i,j}.$$
Constraints:(3)$$\left\{\begin{array}{c} c_{i,j}=f\times d_{i,j},\\ \sum_{i=1}^{n-m}x_{i,j}=-B_{j},\\ \sum_{j=1}^{m}x_{i,j}⩽B_{i},\\ x_{i,j},c_{i,j}⩾0, \end{array}\right.$$
where *T* denotes the total cost of grain trade (unit: US dollar), *m* is the number of grain-deficit countries, n−m is the number of grain-rich countries, $x_{i,j}$, $c_{i,j}$ and $d_{i,j}$ represent the grain trade volume (unit: tonne), transportation cost per unit of grain (unit: US dollar/tonne) and geographical distance from grain-rich country *i* to grain-deficit country *j* (unit: km), respectively, and *f* is the transportation cost per unit of distance and unit of grain (unit: US dollar/km/tonne). In the constraints, $\sum_{i=1}^{n-m}x_{i,j}=-B_{j}$ reflects that the total amount of grain supplied by other countries to country *j* is exactly equal to the grain deficit in country *j*, and $\sum_{j=1}^{m}x_{i,j}⩽B_{i}$ indicates that the total amount of grain supplied by country *i* to other grain-deficit countries should not exceed its own surplus.

Furthermore, the opportunity cost of grain is introduced into Model I and improved to construct a linear optimization Model II with the objective of minimizing the total cost (transportation cost and opportunity cost).

Model II: Linear optimization model with consideration of transportation cost and opportunity cost

Objective function:(4)$$min\;Q=\sum_{i=1}^{n-m}\sum_{j=1}^{m}x_{i,j}\times b_{i,j}.$$
Constraints:(5)$$\left\{\begin{array}{c} b_{i,j}=c_{i,j}+\left(\frac{1}{p_{i}}-\frac{1}{p_{j}}\right),\\ \sum_{i=1}^{n-m}x_{i,j}=-B_{j},\\ \sum_{j=1}^{m}x_{i,j}⩽B_{i},\\ x_{i,j},c_{i,j}⩾0, \end{array}\right.$$
where *Q* denotes the total cost of grain trade with consideration of transportation cost and opportunity cost (unit: US dollar), $b_{i,j}$ is the sum of transportation cost and opportunity cost per unit of grain (unit: US dollar/tonne), and $\frac{1}{p_{i}}$ is the industrial added value generated by the consumption of unit of grain in country *i* (unit: US dollar/tonne). If $\frac{1}{p_{i}}-\frac{1}{p_{j}}>0$, grain trade flows from country *i* with higher grain industrial productivity to country *j* with lower grain industrial productivity; otherwise, grain trade flows in the opposite direction. Opportunity costs in trade are seen from a comparative advantage perspective and not from an absolute advantage. The opportunity cost of grain in a country is assessed in relative values rather than absolute values. Without relying on an accurate monetary assessment, opportunity costs are used in this study to reflect the variability in value added for industrial uses of grain in different countries. From the Food Outlook 2018 [42,43], it is found that the value of *f* is between 0 and 1. For the value of *f* in the model, 10,000 grain trade matrices were obtained by iterating 10,000 times with a step size of 0.0001. By comparing each of the two matrices, it was found that each element of the matrix has an error of 0. The results for Model I and Model II were the same. In other words, the value of *f* does not affect the optimization results of the two models. To simplify the calculation, f=1 is set in this work.

In terms of the practical implications of grain trade, the two models described above are based on the following three reasonable assumptions [34,44]:(1)The market is perfectly competitive and circular, which makes grain a homogeneous product;(2)The total supply of grain is greater than the total demand;(3)Trade flows are supplied from grain-rich countries to grain-deficit countries.

### 2.3. Benefit Analysis

#### 2.3.1. Trade Cost

For the transportation costs involved in Model I and Model II, the variability of transportation costs per unit of grain per unit distance in different countries is not considered in this study. Compared with Model I, the economic benefits due to the regional differences in the industrial added value of grain are analyzed in Model II. The trade cost of grain from country *i* to country *j* can be expressed as Equation (Equation 6):(6)$$E_{i,j}=t_{i,j}\times x_{i,j},$$
where $t_{i,j}$ reflects the trade cost per unit of grain, $t_{i,j}=c_{i,j}$ in Model I and $t_{i,j}=b_{i,j}$ in Model II. A positive economic impact occurs when grain flows from countries with low industrial value added to countries with high industrial value added; otherwise, economic benefits are reduced.

#### 2.3.2. Resource Benefit

In this paper, the virtual water in grain crops *c* (*c* = 1, 2, 3, and 4 represent rice, wheat, maize, and soybean, respectively) is the total water consumed by evapotranspiration during the growth and development of crops. First, the water requirement $ET_{0}$ per unit area of the reference crop is calculated using the Penman–Monteith formula, and the annual water requirement $W_{i}^{c}$ per unit area of the corresponding crop *c* is obtained by adjusting the crop coefficient $k_{c}$ [45]. Combining the harvested area $A_{i}^{c}$ of crop *c* in country *i* and the total annual production $P_{i}$ of the four crops, the virtual water content $W_{i}$ per unit of grain in country *i* can be obtained, with the expression shown in Equation (Equation 7):(7)$$W_{i}=\frac{\sum_{c=1}^{4}W_{i}^{c}\times A_{i}^{c}}{P_{i}},$$
where $W_{i}^{c}=k_{c}\times ET_{0}$, $ET_{0}=\frac{0.408\cdot \Delta \cdot \left(R_{n}-G\right)+\gamma \cdot \frac{900}{T+293}\cdot u_{2}\cdot \left(e_{s}-e_{a}\right)}{\Delta +\gamma \cdot (1+0.34u_{2})}$, $R_{n}$ is the net radiation at the crop surface, *G* is the soil heat flux density, $e_{s}$ is the saturation vapor pressure, $e_{a}$ is the actual vapor pressure, $e_{s}-e_{a}$ is the saturation vapor pressure deficit, Δ is the slope vapor pressure curve, γ is the psychrometric constant, and *T* and $u_{2}$ are the mean of the daily maximum and minimum temperatures and wind speed at 2 m height, respectively. Combining Equations (Equation 1) and (Equation 7), the surplus (or deficit) of country *i*’s grain virtual water can be obtained:(8)$$S_{i}=B_{i}\times W_{i}.$$
Furthermore, the virtual water flow $VW_{i,j}$ from grain-rich country *i* to grain-deficit country *j* can be obtained (Equation 9):
(9)$$VW_{i,j}=x_{i,j}\times W_{i},$$
where the virtual water export volume for grain-rich country *i* is $VW_{i}^{out}=\sum_{j=1}^{m}VW_{i,j}$ and the virtual water import volume for grain-deficit country *j* is $VW_{j}^{in}=\sum_{i=1}^{n-m}VW_{i,j}$.

Water resource benefits arise when virtual water implicit in grain trade flows from countries with high water productivity to those with low water productivity, while water inefficiencies arise when virtual water flows are diverted in the opposite direction. Water productivity can be determined by the virtual water content in the grain. The higher the virtual water content in the grain, the lower the water productivity [46,47]. The water resource benefit $R_{i,j}$ resulting from the virtual water flow from country *i* to country *j* is thus calculated as follows:(10)$$R_{i,j}=\left(W_{i}-W_{j}\right)\times x_{i,j},$$
where $W_{i}$ and $W_{j}$ denote the virtual water content per unit of grain output in outflow country *i* and inflow country *j*, respectively.

In this paper, the water stress index (WS) is used to reflect the environmental stress of water resources and to indicate the scarcity of water resources in different regions. WS is defined as the ratio of annual water withdrawal to renewable water resources in a region [48], and its value ranges from 0 to 1 [49]. A higher value of WS represents a higher scarcity of water resources in the region, and it has a greater potential impact on the environment, human health, and economic development.

## 3. Results

### 3.1. Current Situation of Grain Supply and Demand

In 2018, countries along the B&R produced a total of about 1.54 billion tonnes of grain and consumed 1.51 billion tonnes. This indicates that under the condition of full circulation in 2018, the grain flow can meet the grain demand of countries along the B&R. According to Equation (Equation 1), 19 grain-rich countries and 21 grain-deficit countries can be obtained. The 19 grain-rich countries include Cambodia, Czech Republic, France, Hungary, India, Indonesia, Kazakhstan, Laos, Latvia, Lithuania, Myanmar, Nepal, Moldova, Romania, Russia, Slovakia, Thailand, Ukraine, and Vietnam. The 21 grain-deficit countries include Albania, Armenia, Belarus, China, Egypt, Estonia, Georgia, Iran, Israel, Jordan, Kyrgyzstan, Lebanon, Malaysia, Mongolia, Poland, Slovenia, Sri Lanka, Tajikistan, Turkey, Uzbekistan, and Zimbabwe. From Figure 1, it can be seen that the grain-rich countries are mainly distributed in Eastern Europe and Southeast Asia. European countries such as Ukraine, Russia, Romania and France have abundant land resources, low water pressure and low population density, which make these countries suitable for large-scale grain cultivation and grain export. Of these, Ukraine and Russia are the largest grain-rich countries, with a surplus of about 40% of the total. The climates of south and southeast Asian countries such as Indonesia, India and Thailand are suitable for grain cultivation, and they are the main suppliers of grain trade. Countries such as Egypt, Iran, and Turkey have arid climates and little precipitation, and they are the main countries with grain deficits. In addition, although China is a large grain-producing country, due to its large population and unbalanced agricultural structure, it still needs to import large amounts of grains. The grain deficit of the above four countries accounts for about 75% of the total deficit, and they are the main demand countries in the grain trade.

Equation (Equation 8) is combined with the WS to analyze the relationship between the actual grain balance, virtual water balance, and the water scarcity in each country, as shown in Figure 2. From Figure 2a, it can be found that on the demand side of grains, the countries of China, Malaysia, Egypt, Iran and Israel have negative values of $B_{i}$ and $S_{i}$. That is to say, there are grains, with the transfer of virtual water into these countries. Countries such as Malaysia, Poland and Slovenia are grain-deficit countries and need to import large amounts of grains, leading to large inflows of virtual water. However, these countries have relatively small WS values (as shown in Figure 2b) and relatively abundant water resources, which undoubtedly increases the pressure of water scarcity in the upstream countries of the supply chain to a certain extent. Countries such as Uzbekistan, Egypt, Jordan, Israel and Sri Lanka have relatively large WS values (as shown in Figure 2b), indicating that these countries are short of water resources. It is wise for them to import grains rich in virtual water to alleviate water shortages. On the grain supply side, $B_{i}$ and $S_{i}$ of Thailand, India, Ukraine, Russia and Cambodia are greater than 0, indicating that these countries need to export grain, which is accompanied by the outflow of virtual water. In particular, the WS values of Thailand, Russia, and Cambodia are relatively small (as shown in Figure 2b), and these countries are relatively abundant in water resources, which alleviates the pressure on water resources in the downstream countries of the supply chain to a certain extent. However, Indonesia has a large WS value and transfers a large amount of virtual water outward through grain exports, which further exacerbates its own water stress. In general, in the regional dimension, there is a contradiction between the virtual water flow and the real water flow distribution.

### 3.2. The Optimal Configuration of Grain Trade with Consideration of Transportation Cost

#### 3.2.1. Structure Features

By solving the optimization Model I, the optimal grain supply and demand configuration is obtained, including the amount and direction of grain transfer between countries. The visualization of the optimization results on the map with the objective of minimizing transportation costs is shown in Figure 3a. It can be found that the grain trade pattern is characterized by geographical proximity in space. Each country follows the principle of proximity to distance to select trading partners. Figure 3b,c show the distribution of grain exports and imports, respectively. It can be found that the grain trade volume shows a long-tail feature. With the power function, the fitted equations are $y=38.2x^{-0.89}$ with correlation coefficient $R^{2}=0.94$ and $y=76.3x^{-1.76}$ with correlation coefficient $R^{2}=0.99$, which are extremely significant. This indicates that the grain transfer amount in the optimization result of Model I obeys a typical long-tailed distribution. There are relatively few countries with high trade volumes, and the grain trade has scale-free characteristics.

#### 3.2.2. Benefit Analysis

The optimization results are further analyzed from two aspects of economic cost and resource benefit. According to Equation (Equation 6), the trade cost of grains in the B&R countries can be obtained, as shown in Table 2. It can be found that on the supply side, Russia spends the most on trade at around 137 billion dollars, which is followed by Ukraine and Romania at 117 billion US dollars and 41 billion US dollars, respectively. On the demand side, China has the highest transportation cost at about 345 billion US dollars, which is followed by Egypt and Iran at 30 billion US dollars and 23 billion US dollars, respectively. Overall, the optimal grain trade configuration obtained with the objective of minimizing transportation costs corresponds to a total trade cost of about 466 billion US dollars, which is close to 1/4 of the total agricultural output.

Figure 4a shows the grain trade flows of major countries. The performance of different grain trade flows is discussed below from the perspective of supply and demand, respectively. On the supply side, the largest grain exporter was Ukraine with 35.63 million tonnes, which is followed by Russia and Romania with 23.78 million tonnes and 20.65 million tonnes, respectively. The grain supplies of the above countries accounted for more than half of the total grain trade volume, which was mainly transferred to countries such as China, Egypt and Iran, as shown in Figure 4a. On the demand side, China, Egypt and Iran are the major grain importers. They mainly import about 70% of the total trade volume from countries such as Ukraine, Russia and Romania. In grain trade, a large amount of water resources flow between regions with grain as the carrier. This highlights the need to focus on grains to address water shortages in B&R countries. According to Equation (Equation 9), the total amount of virtual water embodied in grain trade is about $630\times 10^{9}$ cubic meters, accounting for about 4% of the total renewable water resources in the B&R countries. Figure 4b illustrates the main grain virtual water transfer flows. It can be found that on the supply side, the largest virtual water outflow for grains is Thailand with $91\times 10^{9}$ cubic meters, which is followed by India, Cambodia, Indonesia and Russia, with the above countries accounting for about 60% of the total virtual water trade. The grain virtual water in these countries mainly flows into China, Malaysia and other countries. The virtual water content of grain can reflect the water productivity. The higher the virtual water content of grain, the lower the water productivity. As a result, the virtual water content of grains varies in different countries, leading to the occurrence of water savings or inefficiencies. Figure 4c compares the changes in the ranking of grain transfers and virtual water transfers. It can be observed that the two rankings are quite different for most countries. Grain transfers from countries such as Thailand, Cambodia and Myanmar to China are not ranked high, while virtual water transfers are far ahead. This suggests that water productivity in these grain-exporting countries is low, leading to the occurrence of water inefficiencies. Conversely, grain transfers from Romania to Egypt and Ukraine to Iran rank 2nd and 3rd, respectively, while the corresponding virtual water transfers rank 7th and 10th. This reflects the high water productivity of grain-exporting countries, resulting in water savings.

Furthermore, Table 3 lists the water benefits in grain trade in BRI countries with consideration of transportation cost. In terms of water savings along the grain transfer path, the trade from Ukraine to Iran achieved the largest water savings, reaching $46\times 10^{9}$ cubic meters, which was followed by Romania to Egypt ($31\times 10^{9}$ cubic meters) and Indonesia to Malaysia ($23\times 10^{9}$ cubic meters). The top three trade flows in grain transfer paths where water inefficiency occurs are from Thailand, India and Cambodia, which are all destined to China. The resulting water resources losses are $72\times 10^{9}$ cubic meters, $56\times 10^{9}$ cubic meters and $55\times 10^{9}$ cubic meters, respectively, which together account for about 70% of the total water resources losses. This is consistent with the results presented in Figure 4c. Overall, in the optimization results considering transportation costs, the water loss caused by grain trade in the B&R countries is about $72\times 10^{9}$ cubic meters. The negative water resource benefits of grain trade are mainly because the virtual water embodied in grains mostly flows from countries with low water productivity to those with high water productivity. Furthermore, Figure 5a,b show the contribution of grain exporters and importers to water savings, respectively. As shown in Figure 5a, on the grain supply side, Ukraine and Romania have made the largest contributions to water savings. They save more than 80% of the total water resources saved in the outflow area. On the demand side, the top three countries contributing to water saving are Iran, Egypt and Malaysia. Their grain import trade saves more than half of the total water savings in the inflow area.

### 3.3. The Optimal Configuration of Grain Trade with Consideration of Transportation Cost and Opportunity Cost

#### 3.3.1. Structure Features

The optimal grain supply and demand configuration for the countries along the B&R with consideration of transportation cost and opportunity cost is obtained by solving the optimization Model II. The visualization of the optimization results on the map is presented in Figure 6a. It can be found that the grain trade pattern has obvious geographical clustering characteristics in space. In other words, spatial location is an important influencing factor when countries choose their trading partners. Figure 6b,c show the distribution of grain exports and imports in the B&R countries, respectively. By fitting the data, the fitting functions and coefficients of determination obtained are $y=44.6x^{-0.98}$ ($R^{2}=0.96$) and $y=76.3x^{-1.76}$ ($R^{2}=0.99$), respectively, which show a good fit. The obtained optimization results of Model II are consistent with those of Model I. The grain transfers of the B&R countries conform to the scale-free characteristics of the power index distribution. However, there is a phenomenon that the power index is relatively low. This is manifested as a large amount of grain transfers in a few countries, while there is only a small amount in most countries, like a long tail.

#### 3.3.2. Benefit Analysis

Next, the optimization results of Model II are analyzed from the perspective of economic cost and resource benefit. The total cost of grain trade in B&R countries obtained from Equation (Equation 6) is shown in Table 4. It is worth mentioning that if the total trade cost is negative, this means that the trade flow is directed from countries with higher industrial added value of grains to those with lower industrial added value. The economic value generated in the process not only covers the transportation costs but also generates additional economic benefits. In terms of economic benefits from opportunity cost in grain trade, Romania has the highest on the supply side at around 70 trillion US dollars, which is followed by Cambodia and Ukraine at nearly 27 trillion US dollars and 13 trillion US dollars, respectively. In contrast, Russia’s trade costs are positive, which shows that the economic value generated by opportunity cost does not cover the transportation costs, so that no additonal economic benefits are generated. On the demand side, Lebanon has the highest economic benefit at about 48 trillion US dollars, which is followed by Malaysia (30 trillion US dollars) and Israel (17 trillion US dollars). Overall, with the objective of minimizing the total cost of grain trade, the optimized trade configuration corresponds to an economic benefit of about 130 trillion US dollars, which is nearly 65 times the total agricultural output of the B&R countries studied. Therefore, considering the opportunity cost in grain trade has a positive effect on improving the economic development level of the countries along the B&R. Furthermore, Figure 7 compares the trade costs of the B&R countries in the two optimization results. Compared with Model I, most countries in Model II have lower trade costs, and some have additional economic gains. However, in the optimal configuration of Model II, Lithuania exports grain to countries with lower industrial added value, while Belarus and Zimbabwe mainly import grain from countries with higher industrial added value, which leads to an increase in their total trade costs.

Figure 8a shows the main grain trade flows among the major countries. Similar to the optimization results with consideration of transport cost, on the supply side, the three largest grain-exporting countries are Ukraine, Russia and Romania, which supply almost 60% of the total grain trade. Among them, Ukraine exported nearly 44 million tonnes, which is more than the other two countries combined. On the demand side, China, Egypt and Iran remain major grain importers. Their imports account for about 70% of the total trade volume, mainly from Ukraine and Russia. The water resource benefits resulting from the optimal grain trade structure obtained by Model II are considered below. The virtual water volume in grain trade is reduced by about $100\times 10^{9}$ cubic meters compared to that of Model I, and its share of the total renewable water resources of the Belt and Road countries is reduced from 4% to 3%. Figure 8b illustrates the major grain virtual water transfer flows. Similar to the results of Model I, the grain virtual water flows are mainly transferred from countries such as Thailand, India, Ukraine and Cambodia to countries such as China and Malaysia. It can be found that the two rankings of most countries are relatively close. There are fewer transfer paths, such as from Thailand to Malaysia, Laos to China, etc., with grain transfer ranked 26th and 17th, respectively, while virtual water transfer ranked 10th and 8th. This reflects the occurrence of water inefficiencies due to lower water productivity in grain-exporting countries.

Table 5 lists the water resources benefits generated by the optimal grain trade structure of the B&R countries considering transportation costs and opportunity costs. In terms of water savings along the grain transfer path, the trade flow from Ukraine to Iran ranks first, with water savings close to $30\times 10^{9}$ cubic meters, which is followed by Ukraine to China ($25\times 10^{9}$ cubic meters) and Romania to Israel ($17\times 10^{9}$ cubic meters). Among the grain transfer paths where water inefficiency occurs, the top three trade flows originate from Thailand, India and Myanmar, with the main destination being China. The resulting water losses are $62\times 10^{9}$ cubic meters, $52\times 10^{9}$ cubic meters and $23\times 10^{9}$ cubic meters, respectively. In general, in the optimization results of Model II, the grain trade in the B&R countries has produced positive water resources benefits, with water savings of nearly $64\times 10^{9}$ cubic meters. This suggests that grain flows are mainly from countries with high water productivity to those with low water productivity. Furthermore, Figure 9a,b depict the contribution of grain exporters and importers to water savings, respectively. It can be found that on the supply side, the top three countries with water-saving contributions include Romania, Ukraine and France. The water resources saved account for more than 90% of the total saving in the outflow area. On the demand side, Iran, Egypt and Israel contribute the most to water savings, accounting for approximately 60% of the total water savings in the inflow area.

## 4. Conclusions and Policy Implications

The B&R Initiative has built a platform for China and related countries to smooth trade and promote the development of world economic and trade cooperation. With the steady increase in the scale of trade in recent years, the environmental concerns associated with the B&R initiative cannot be ignored. Water scarcity has become an important environmental issue facing the world today. Some studies have revealed at the regional level that the pattern of international grain virtual water flows exhibits a north–south pattern, which to some extent exacerbates the imbalance of regional resource allocation [30,31,32]. In addition, this trade pattern is not conducive to balancing grain supply and demand across regions. For grain-exporting countries or regions where water resources are scarce, the lack of sustainability of agricultural production and water resources can exacerbate the pressure on local water resources, further threatening grain production. On the grain import side, if the embodied water resources in grain trade are not effectively utilized, it will result in an indirect waste of resources, which constrains sustainable economic and social development. In response to the current mismatch between the spatial distribution of grain production and water resources, this study designs the grain trade optimization model for countries along the Belt and Road to reconfigure the structure of grain supply and demand and achieve a rational allocation of resources, which is meaningful and challenging. In this study, the total grain trade cost is divided into two parts: transportation cost and opportunity cost. A linear optimization model is constructed to explore the optimal configuration of grain trade in B&R Initiative countries with the objectives of minimizing transportation cost and total trade cost, respectively. The results are as follows.

(1)Current situation of grain trade

Grain resources are highly concentrated. Grain-rich countries are mainly located in Eastern Europe and Southeast Asia. Ukraine and Russia are the largest grain-rich countries, with surpluses of about 40% of the total. China, Egypt, Iran and Turkey are the main demand countries in the grain trade, and their grain deficit accounts for about 75% of the total deficit. Comparing the relationship between grain balance, embodied virtual water balance, and water scarcity in various countries, it is found that on the demand side, countries with relatively abundant water resources such as Malaysia and Poland have introduced a large amount of virtual water through grain imports, which undoubtedly increases the pressure on countries upstream in the supply chain, especially water-scarce countries. On the supply side, Indonesia faces water shortages, but it transfers a large amount of virtual water through grain exports, which further exacerbates its own water scarcity. In general, there is a contradiction between the grain virtual water flow pattern and the actual water flow distribution in the countries along the B&R Initiative.

(2)Selection preference features

Both optimization results reveal that trading countries follow geographic proximity and long-tail characteristics in choosing partners. This is due to the fact that geographical proximity can reduce the transportation and communication costs of countries and avoid the increase of transaction cost caused by information asymmetry. Therefore, trading countries are more inclined to conduct trade with neighboring countries in order to obtain the advantages brought by geographical proximity. In addition, the grain trade of B&R Initiative countries has scale-free characteristics, indicating that trading countries have the characteristics of selection preference and tend to cooperate with countries with high trade volume to ensure the security of grain supply.

(3)Optimization of grain trade with consideration of transportation cost

The grain supply and demand structure obtained with the objective of minimizing transportation costs corresponds to a trade cost of about 466 billion US dollars, which is close to 1/4 of the total agricultural output. The corresponding loss of water resources is about $72\times 10^{9}$ cubic meters, reflecting that most of the virtual water embodied in grain trade flows from countries with low water productivity to those with high water productivity, resulting in an inefficient use of water resources.

(4)Optimization of grain trade with consideration of transportation cost and opportunity cost

Based on the optimization model with the objective of minimizing the transportation cost, the economic value of industrial grain is introduced as the opportunity cost, and then, the linear optimization model with the objective of minimizing the total trade cost (transportation cost and opportunity cost) is developed to reconstruct the grain supply and demand relationship. The results show that the introduction of opportunity cost constrains the grain trade flow from the outflow area with low industrial added value to the inflow area with high industrial added value, which has a positive economic impact. The economic value generated not only covers the transportation cost but also has an additional economic benefit of about 130 trillion US dollars. Furthermore, the analysis of grain virtual water trade shows that the optimization results also bring positive water resources benefits, saving about $28\times 10^{9}$ cubic meters of water resources.

Based on these findings, the following important insights can be drawn.

(1) Given the priority of some countries in grain trade, they have great potential to reshape the structure of grain supply and demand. In particular, China should take full advantage of the development opportunities brought by the BRI to strengthen extensive cooperation with neighboring grain-rich countries, such as Russia and India. Then, it radiates to the main exporting countries along the route, such as the Czech Republic, Romania, Kazakhstan and other countries. Finally, a mutually beneficial and win–win development pattern will be formed.

(2) The optimal grain trade configuration with consideration of opportunity cost yields significant economic and water resources benefits. Opportunity costs can be a means of improving trade structures and managing water resources. Noting this, the B&R economies should consider not only the natural endowments of the region but also the opportunity cost of grain products when reshaping the grain trade structure.

(3) Countries on the grain supply side can improve water productivity and vigorously develop grain production by improving crop varieties, enhancing soil fertility, and introducing water-saving technologies. Countries on the grain demand side should focus on developing their economies and improving the industrial and economic value of grain products.

(4) The optimized grain supply and demand structure with the introduction of opportunity cost has brought significant economic benefits to the B&R economies. However, the shortcoming is that some countries suffer economic losses due to the increase in trade costs compared to optimization results with consideration of transportation cost. For example, Lithuania exports grain to countries with lower industrial added value, while Belarus and Zimbabwe mainly import grain from countries with higher industrial added value, which leads to an increase in their total trade costs. Therefore, a regional cooperation organization should be established to explore the compensation mechanism for these countries and to conduct coupled management of physical–virtual water to ensure the sustainable and smooth flow of grains in B&R countries.

The findings have implications for grain security and water management, but there are still some shortcomings. Firstly, it is one-sided to evaluate the security of grain supply and demand only from the perspective of water resources, because the optimization of the grain trade structure also involves virtual land, virtual labor and other determinants. Benefiting from the results of the research at hand, further research will consider how to account for the virtual land embodied in grain trade and how to assess the ecological benefits resulting from the conversion of land resources saved due to the optimized structure of grain trade into ecological land. In addition, with the increased granularity of MRIO tables, it is necessary and meaningful to obtain the optimal configuration of grain trade in B&R countries based on the input–output approach and to compare and analyze the results with those in the study.

## Figures and Tables

**Figure 1 entropy-24-01667-f001:**
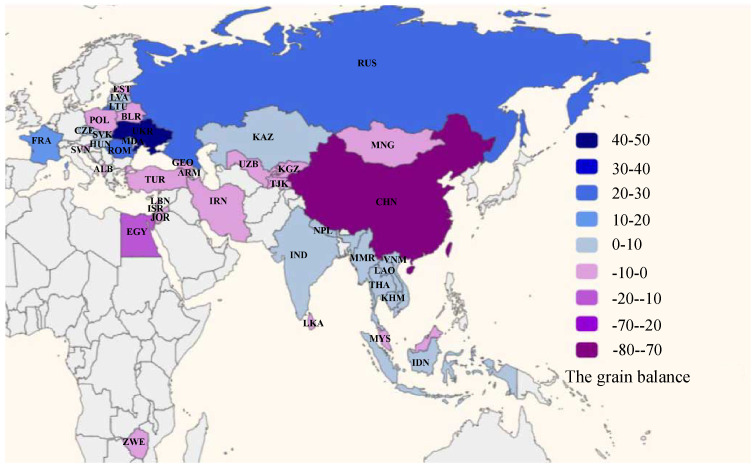
The current status of surpluses and deficits of grain in B&R countries. Notes: Negative values indicate deficits (purple), positive values indicate surpluses (blue) (unit: $10^{6}$ tonnes).

**Figure 2 entropy-24-01667-f002:**
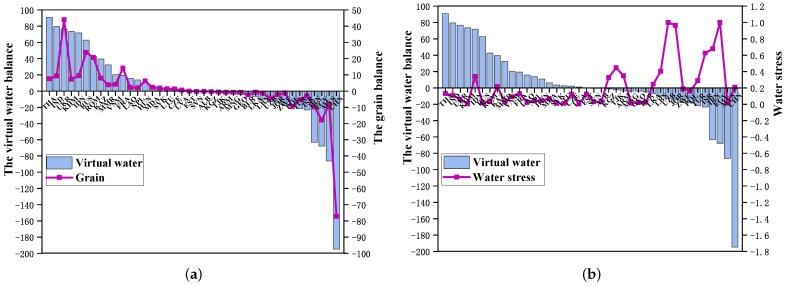
(**a**) Relationship between grain balance (unit: $10^{6}$ tonnes) and virtual water balance (unit: $10^{9}$ cubic meters). (**b**) Relationship between water stress and virtual water balance.

**Figure 3 entropy-24-01667-f003:**
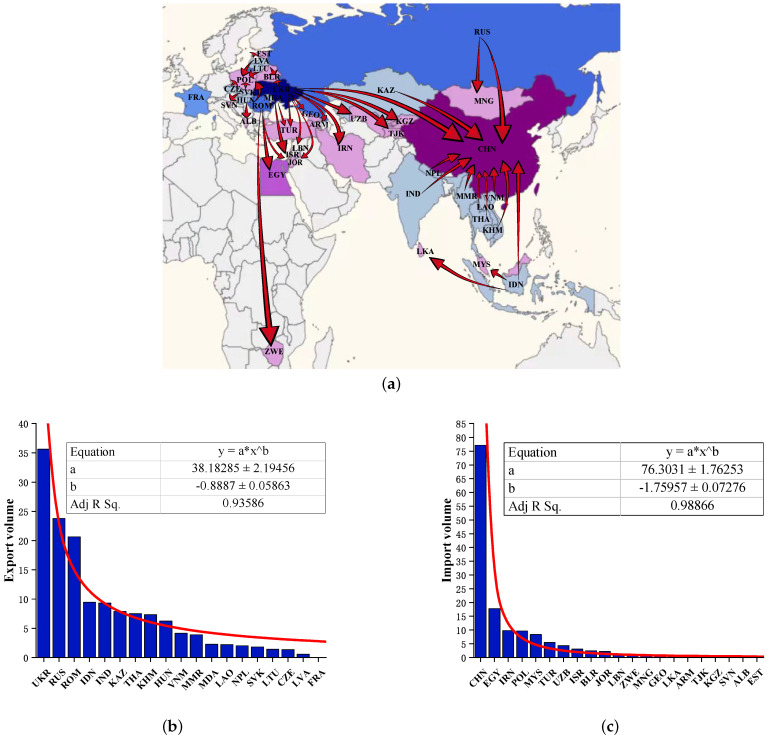
(**a**) Visualization of the optimization results of Model I on the map. (**b**) Distribution of grain trade exports (unit: $10^{6}$ tonnes). (**c**) Distribution of grain trade imports (unit: $10^{6}$ tonnes).

**Figure 4 entropy-24-01667-f004:**
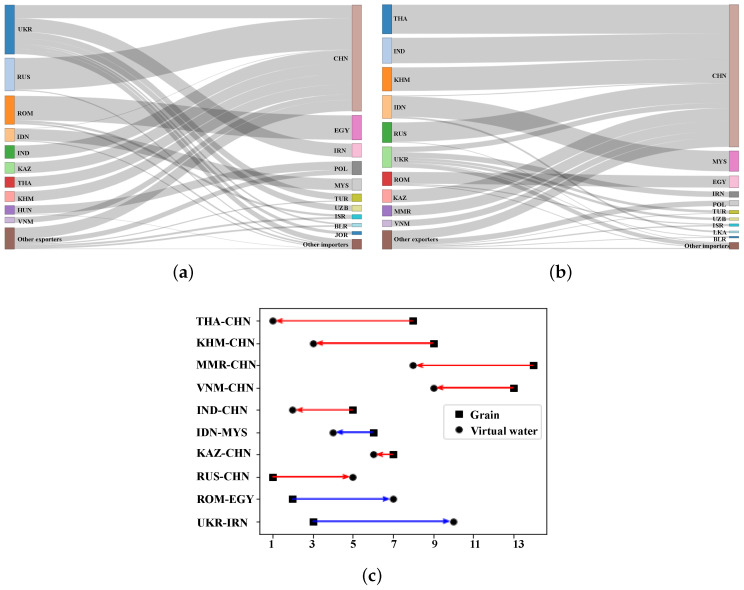
(**a**) The main grain transfer flows of Model I. (**b**) The main virtual water transfer flows of Model I. (**c**) Ranking change from grain transfers to virtual water transfers of Model I. Notes: Blue arrows indicate grain flow with water-saving benefits, red arrows indicate water-inefficient grain flow.

**Figure 5 entropy-24-01667-f005:**
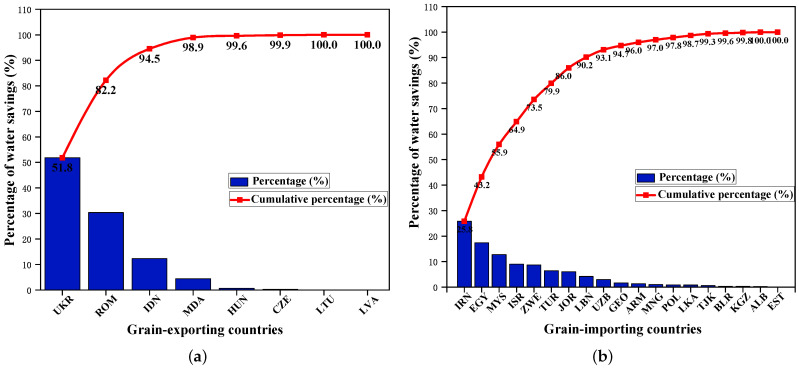
Distribution of water savings in grain trade in B&R countries with consideration of transportation cost. (**a**) For exporting countries. (**b**) For importing countries.

**Figure 6 entropy-24-01667-f006:**
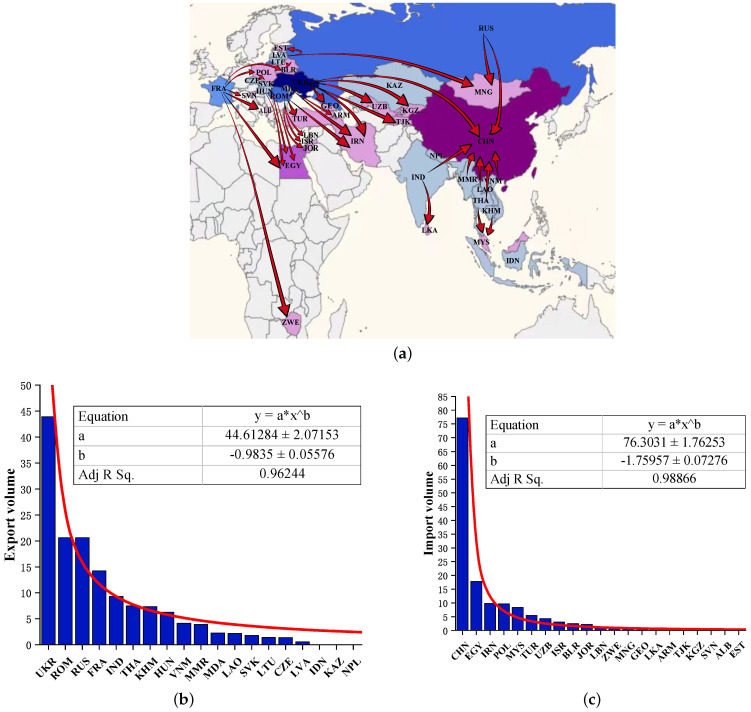
(**a**) Visualization of the optimization results of Model II on the map. (**b**) Distribution of grain trade exports (unit: $10^{6}$ tonnes). (**c**) Distribution of grain trade imports (unit: $10^{6}$ tonnes).

**Figure 7 entropy-24-01667-f007:**
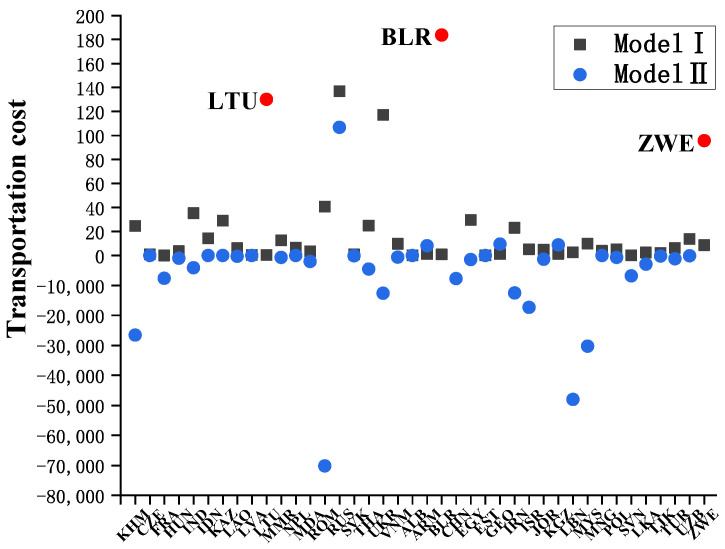
Comparison of the trade costs of B&R countries in the two optimization results.

**Figure 8 entropy-24-01667-f008:**
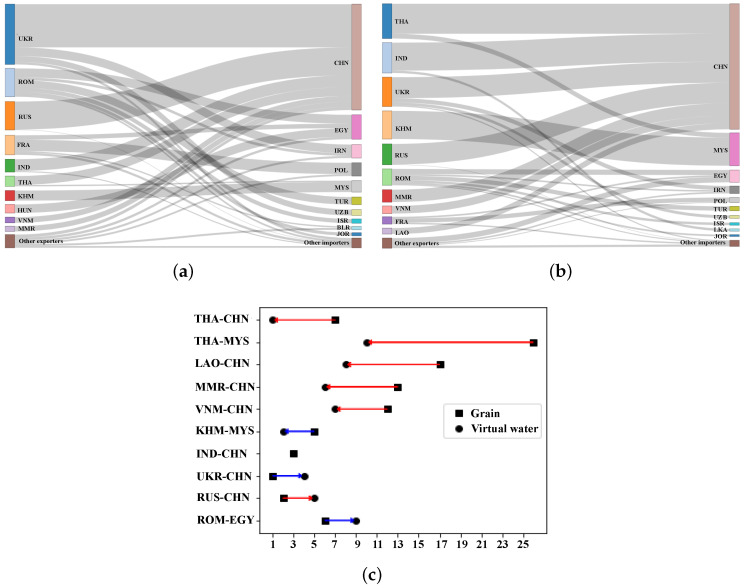
(**a**) The main grain transfer flows of Model II. (**b**) The main virtual water transfer flows of Model II. (**c**) Ranking change from grain transfers to virtual water transfers of Model II. Notes: Blue arrows indicate grain flow with water-saving benefits, red arrows indicate water-inefficient grain flow.

**Figure 9 entropy-24-01667-f009:**
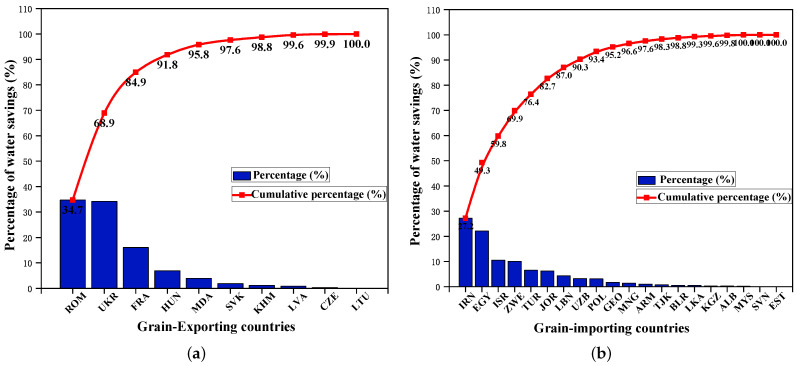
Distribution of water savings in grain trade in B&R countries with consideration of transportation cost and opportunity cost. (**a**) For exporting countries. (**b**) For importing countries.

**Table 1 entropy-24-01667-t001:** Accounting details of data.

Categories		Accounting Methods
Ration consumption	Household consumption	Per capita grain consumption × Population
	External household consumption	Household consumption × 16%
Industrial consumption	Alcohol	The conversion rate 1:3
	Alcoholic beverages	The conversion rate 1:2.3
	Fermented beverages	The conversion rate 1:0.172
	Other industrial consumption	The above industrial consumption × 25%
Feed consumption	Beef	The conversion rate 1:1.7
	Mutton	The conversion rate 1:1.7
	Pork	The conversion rate 1:3.5
	Poultry	The conversion rate 1:1.7
	Fish	The conversion rate 1:0.9
	Eggs	The conversion rate 1:2.2
Seed consumption	Rice	The per unit area seed consumption 75 kg/ha
	Maize	The per unit area seed consumption 75 kg/ha
	Soybean	The per unit area seed consumption 75 kg/ha
	Wheat	The per unit area seed consumption 225 kg/ha
Grain loss	Inventory loss	Grain production × 2%
	Transportation loss	(Ration consumption + Industrial consumption + Feed consumption) × 4%
	Processing loss	(Ration consumption + Industrial consumption + Feed consumption) × 5%

**Table 2 entropy-24-01667-t002:** The grain trade costs of the B&R countries with consideration of transportation cost.

	Importers	ALB	ARM	BLR	CHN	EGY	EST	GEO	IRN	ISR	JOR	KGZ	LBN	MYS	MNG	POL	SVN	LKA	TJK	TUR	UZB	ZWE	Total
Exporters																							
KHM					24.579																		24.579
CZE																0.704							0.704
FRA																							0.000
HUN		0.132														3.311							3.443
IND					35.259																		35.259
IDN					1.847									9.817				2.443					14.108
KAZ					28.884																		28.884
LAO					6.076																		6.076
LVA							0.007									0.297							0.304
LTU				0.210												0.072							0.282
MMR					12.578																		12.578
NPL					6.366																		6.366
MDA										2.531										0.689			3.220
ROM						29.579				2.556												8.463	40.598
RUS					133.174										3.709								136.882
SVK																0.804	0.089						0.893
THA					24.782																		24.782
UKR			1.118	0.528	61.579			1.129	23.090		4.696	1.332	2.499						2.025	5.516	13.583		117.094
VNM					9.690																		9.690
Total		0.132	1.118	0.738	344.815	29.579	0.007	1.129	23.090	5.087	4.696	1.332	2.499	9.817	3.709	5.188	0.089	2.443	2.025	6.205	13.583	8.463	465.744

unit: 10^9^ US dollars.

**Table 3 entropy-24-01667-t003:** Water resource benefits of grain trade in B&R countries with consideration of transportation cost.

	Importers	ALB	ARM	BLR	CHN	EGY	EST	GEO	IRN	ISR	JOR	KGZ	LBN	MYS	MNG	POL	SVN	LKA	TJK	TUR	UZB	ZWE	Total
Exporters																							
KHM					55.017																		55.017
CZE																−0.510							−0.510
FRA																							0.000
HUN		−0.315														−0.958							−1.272
IND					55.973																		55.973
IDN					1.791									−22.790				−1.495					−22.494
KAZ					19.506																		19.506
LAO					10.125																		10.125
LVA							−0.005									−0.061							−0.066
LTU				−0.140												0.001							−0.139
MMR					22.570																		22.570
NPL					8.546																		8.546
MDA										−6.999										−0.993			−7.992
ROM						−30.953				−8.964												−15.391	−55.308
RUS					2.570										−1.776								0.794
SVK																0.013	0.090						0.103
THA					71.873																		71.873
UKR			−2.320	−0.328	−7.423			−2.852	−45.986		−10.760	−0.397	−7.465						−1.161	−10.427	−5.240		−94.360
VNM					9.549																		9.549
Total		−0.315	−2.320	−0.468	250.096	−30.953	−0.005	−2.852	−45.986	−15.963	−10.760	−0.397	−7.465	−22.790	−1.776	−1.514	0.090	−1.495	−1.161	−11.420	−5.240	−15.391	71.914

Notes: Negative numbers represent water savings, and positive numbers represent water inefficiency (unit: 10^9^ cubic metres).

**Table 4 entropy-24-01667-t004:** The grain trade costs of the B&R countries with consideration of transportation cost and opportunity cost.

	Importers	ALB	ARM	BLR	CHN	EGY	EST	GEO	IRN	ISR	JOR	KGZ	LBN	MYS	MNG	POL	SVN	LKA	TJK	TUR	UZB	ZWE	Total
Exporters																							
KHM														−26.554									−26.554
CZE																−0.051							−0.051
FRA		−0.056		0.054		−0.273										−0.635	−6.811					0.096	−7.626
HUN						−0.917																	−0.917
IND					−1.217													−2.930					−4.147
IDN																							0.000
KAZ																							0.000
LAO					−0.310																		−0.310
LVA							−0.004								−0.026								−0.030
LTU				0.130																			0.130
MMR					−0.652																		−0.652
NPL																							0.000
MDA			0.008						−2.009														−2.001
ROM						−0.088			−2.359	−17.317	−1.289		−48.016							−1.177			−70.246
RUS					0.085										0.022								0.107
SVK						−0.149																	−0.149
THA					−0.885									−3.719									−4.604
UKR					−4.160			0.009	−8.135			0.009							−0.261		−0.102		−12.640
VNM					−0.595																		−0.595
Total		−0.056	0.008	0.184	−7.734	−1.427	−0.004	0.009	−12.504	−17.317	−1.289	0.009	−48.016	−30.273	−0.004	−0.686	−6.811	−2.930	−0.261	−1.177	−0.102	0.096	−130.286

Notes: Negative numbers indicate an increase in economic benefit; positive numbers indicate a decrease in economic benefit (unit: 10^12^ US dollars).

**Table 5 entropy-24-01667-t005:** Water resource benefits of grain trade in B&R countries with consideration of transportation cost and opportunity cost.

	Importers	ALB	ARM	BLR	CHN	EGY	EST	GEO	IRN	ISR	JOR	KGZ	LBN	MYS	MNG	POL	SVN	LKA	TJK	TUR	UZB	ZWE	Total
Exporters																							
KHM														−2.099									−2.099
CZE																−0.510							−0.510
FRA		−0.389		−0.689		−7.898										−4.527	−0.069					−16.262	−29.834
HUN						−12.900																	−12.900
IND					51.551													−0.798					50.753
IDN																							0.000
KAZ																							0.000
LAO					10.125																		10.125
LVA							−0.005								−1.632								−1.638
LTU				−0.161																			−0.161
MMR					22.570																		22.570
NPL																							0.000
MDA			−1.606						−5.725														−7.331
ROM						−11.357			−8.703	−16.969	−10.052		−7.045							−10.497			−64.623
RUS					2.279										−0.599								1.680
SVK						−3.426																	−3.426
THA					62.040									1.842									63.882
UKR					−24.533			−2.852	−29.356			−0.397							−1.161		−5.240		−63.539
VNM					9.549																		9.549
Total		−0.389	−1.606	−0.851	133.581	−35.581	−0.005	−2.852	−43.784	−16.969	−10.052	−0.397	−7.045	−0.257	−2.231	−5.036	−0.069	−0.798	−1.161	−10.497	−5.240	−16.262	−27.501

Notes: Negative numbers represent water savings, and positive numbers represent water inefficiency (unit: 10^9^ cubic metres).

## Data Availability

Not applicable.

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
