# Peer review of "Optimization and Benefit Analysis of Grain Trade in Belt and Road Countries"

_entropy, 2022, doi:10.3390/e24111667_

Round 1

Reviewer 1 Report

The paper introduces opportunity costs on the discussion of grain trade efficiency. However, it has many caveats and the contribution is not enough to deserve a publication. ​Nevertheless, it is important to note that this is not my field of work, and there may be some misunderstandings. ​

The model is based on the summation of transport and opportunity costs. However, they are not in the same unit, which is necessary to sum-up them (equation 5). The inclusion of VA/unit as a measure for the opportunity costs is not explained, as well as how "industrial value added" is obtained. Usually, opportunity costs in trade are seen from a Comparative Advantage perspective, and not from an Absolute Advantage. Even if is more expensive in country j than country i, we need to consider the country-specific costs to produce other products.

In terms of the model assumptions (section 2.2), there is no reference arguing that these are the only ones. In fact, even for these ones, references are needed.

Moreover, the f does not affect the Model 1 because it is only multiplying x; here it may be relevant to use the correct value.

Moreover, it argues that it does not use IO tables because of the level of aggregation and because it is not up-to-date. However, the granularity of IO tables has been increasing, and now there are good databases at the crop level. In terms of time, some of them are updated annually, and have data for less than two years ago (see, for example, GLORIA, developed by Lenzen et al., (2022)).

Lenzen, M., Geschke, A., West, J., Fry, J., Malik, A., Giljum, S., Milà i Canals, L., Piñero, P., Lutter, S., Wiedmann, T., Li, M., Sevenster, M., Potočnik, J., Teixeira, I., Van Voore, M., Nansai, K. and Schandl, H. (2022) Implementing the material footprint to measure progress towards Sustainable Development Goals 8 and 12. Nature Sustainability, 5, 157-166. https://doi.org/10.1038/s41893-021-00811-6

Author Response

请参阅附件

Reviewer 2 Report

The paper “Optimization and Benefit Analysis of Grain Trade in Belt and Road Countries“ submitted to Entropy reads as an elegant piece of research. The questions are clear and relevant. The chosen methodological approach is sound. The conduct seems proper.

The research design remains rather basic and straightforward. Linking transportation cost and water consumption is relevant, yet not original. Qualitative aspects are missing. The paper would have benefitted from providing an explicit critical perspective not only to the subject of the research but to the research itself. What can be criticised in the research design and conduct and what can be improved in which way? In addition and in a similar fashion, the paper would benefit from addressing explicitly possible leads for further research building on the results of the research at hand. What are new research questions and possible designs resulting from the research?

Author Response

请参阅附件

Round 2

Reviewer 1 Report

The authors changed the text according to the suggestions, and it is better now.